

# Automated generation of process simulation scenarios from declarative control-flow changes

Daniel Barón-Espitia[1], Marlon Dumas[2] and Oscar González-Rojas[1]

[1] Systems and Computing Engineering Department, Universidad de los Andes, Bogotá, Colombia
[2] University of Tartu, Tartu, Estonia

## ABSTRACT

Business process simulation is an established approach to estimate the potential impact of hypothetical changes on a process, particularly in terms of time and cost-related performance measures. To overcome the complexity associated with manually specifying and fine-tuning simulation models, data-driven simulation (DDS) methods enable users to discover accurate business process simulation models from event logs. However, in the pursuit of accuracy, DDS methods often generate overly complex models. This complexity can hinder analysts when attempting to manually adjust these models to represent what-if scenarios, especially those involving control-flow changes such as activity re-sequencing. This article addresses this limitation by proposing an approach that allows users to specify control-flow changes to a business process simulation model declaratively, and to automate the generation of what-if scenarios. The proposed approach employs a generative deep learning model to produce traces resembling those in the original log while implementing the user-specified control-flow changes. Subsequently, the technique generates a stochastic process model, and uses it as a basis to construct a modified simulation model for what-if analysis. Experiments show that the simulation models generated through this approach replicate the accuracy of models manually created by directly altering the original process model.

## INTRODUCTION

Business process simulation (BPS) refers to the practice of creating a model that mimics the executions of an actual business process, and using this model to analyze the impact of changes on the performance of the process (*Dumas et al., 2018*). BPS is typically used to compare multiple possible changes to a process, also known as *what-if scenarios*, with the goal of selecting and prioritizing improvement opportunities.

The skeleton of a BPS model typically consists of a process model, often represented in the business process model and notation (BPMN) standard. Additionally, a BPS model incorporates various simulation parameters related to the performance of activities (primarily, their processing times or durations), the resources involved in the process and their availability, the branching behavior in the decision points of the process model, and

Corresponding author
Oscar González-Rojas,
o-gonza1@uniandes.edu.co

other components such as the rate at which new cases are created. Once a BPS model is created, a simulator can be employed to generate a number of hypothetical instances of the process (cases) by simulating the execution of each case step-by-step and recording each step. The simulator's output includes simulation logs as well as statistics such as cycle times, average waiting times, and average resource utilization. These statistics can be used to assess the performance of the process under the given parameters.

A recurrent roadblock when applying BPS in practice is the complexity of manually designing simulation models and, especially, manually tuning these simulation models to achieve a suitable level of accuracy in reflecting the observed behavior of the process. To tackle this roadblock, a number of so-called data-driven simulation (DDS) approaches have been proposed (*Rozinat, Mans & van der Aalst, 2009*; *Khodyrev & Popova, 2014*; *Martin, Depaire & Caris, 2016*; *Pourbafrani, van Zelst & van der Aalst, 2020*; *Camargo, Dumas & González-Rojas, 2020*). These approaches automatically construct and fine-tune simulation models from event logs using process mining (PM) techniques. However, a drawback of these approaches is that, in their pursuit of accuracy, they often generate complex simulation models. These models contain a significant number of control-flow transitions and intricate branching and looping structures. This complexity complicates the comprehension of control-flow relations in the simulation models for analysts and hinders their ability to manually adjust them to capture what-if scenarios (*e.g.*, what would happen if we resequence two tasks). Moreover, when analysts modify simulation models with high levels of complexity, they may inadvertently introduce control-flow structures that lead to deadlocks.

Previous work that has demonstrated the feasibility of adapting declarative rules to procedural models (*De Masellis et al., 2017*). This article presents an approach to automate the generation of what-if simulation scenarios based on control-flow changes. This literature-first approach is named Dynamics (Declarative Yielding of Accurate Models for Interactive Control-flow Simulation) and allows for:

C1   Specifying control-flow changes declaratively. Users can specify control-flow changes declaratively, considering two categories of changes: restrictive changes, where the specified control-flow changes restrict the set of possible traces, and enhancing changes, which introduce behavior not allowed by the original process model.

C2   Generating traces that adhere to specified changes. Dynamics uses a deep learning (DL) model learned from the original event log to generate traces that adhere to the declarative constraints. These traces are then used to create a target (deadlock-free) process model.

C3   Automating the generation of simulation scenarios. Dynamics enriches the generated process model with simulation parameters discovered from the log, producing a simulation model that captures the scenario after the control-flow change. This method empowers business users to evaluate the performance implications of such changes in the simulated environment.

An experimental evaluation shows that simulation models generated from declarative specifications replicate the accuracy of manually modified models when estimating

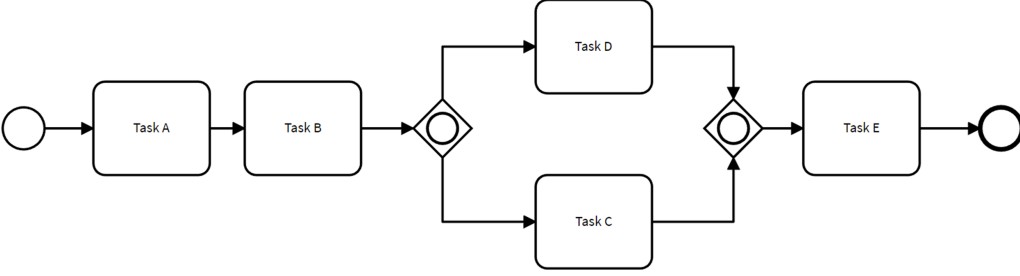

**Figure 1** Example of a simple process model.

performance of process. The first experiment compares the effects of both restrictive and enhancing changes on synthetic logs, while the second experiment examines the effect of restrictive changes on a real-life event log with complex control-flow behaviors. Moreover, user specified changes enhance what-if analysis by allowing the specification of incremental simulation scenarios on a continuous basis. The evaluation delves into the relative accuracy of different types of control-flow changes to represent and validate what-if scenarios.

The article is structured as follows. 'Background and Related Work' discusses methods related to learning and generating models using DDS and DL techniques, as well as specifying declarative rules for procedural models. 'Automatic Generation of Simulation Scenarios with Control-Flow Changes' outlines the proposed approach while 'Evaluation' presents an empirical evaluation. Finally, 'Conclusions and Future Work' draws conclusions and outlines future work.

# BACKGROUND AND RELATED WORK

This section introduces concepts of data-driven process simulation related to the proposed approach and related work.

## Context: process simulation
### Process models and simulation models

A *business process model* represents a set of activities that occur within a process, along with the control-flow dependencies and other types of dependencies between these activities. Business process modeling enables companies to document how a process functions, including its inputs, outputs, decision points, and the roles and responsibilities of the resources involved. It provides organizations with insights to identify inefficiencies, bottlenecks, and opportunities for process improvement (*Aguilar-Savén, 2004*). Figure 1 illustrates a small process model used as a running example.

*Stochastic process models* capture the probabilistic nature of events, activities, and decisions within process models (*Burke et al., 2022*). Specifically, a stochastic process model is a process model enhanced with branching probabilities. These probabilities are attached to the control-flow transitions stemming from decision gateways, determining the likelihood of taking one path over others when a decision gateway is reached. For example,

the process model depicted in Fig. 1 could be enhanced into a stochastic process model by associating a probability with each of the two control-flow transitions stemming from the inclusive decision gateway after Task B.

A *process model* captures how a set of human and automated resources perform activities and interact to deliver a product or a service. Each execution of a process, also known as a *case*, consists of instances of activities, such as handling a customer request or verifying a claim. These activity instances are typically recorded in process-aware information systems (PAIS), which then allow us to extract an audit trail of the execution of the process in question. In These audit trails are called event logs.

An *event log* is a structured collection of event records, containing essential information about the execution of activity instances in a process. For each activity instance within a case, an event log provides a case identifier, activity name, start and end execution timestamps, and additional attributes describing associated data (*e.g.*, resource, channel, information objects, cost, *etc.*). Process mining (PM) techniques utilize event logs as input to elucidate the causal relationships among observed activities in the process and their attributes (*Rozinat, Mans & van der Aalst, 2009*). PM facilitates the discovery of execution paths, referred to as traces, which depict the sequence of activities executed by specific resources for a particular case. This enables descriptive and diagnostic quantitative analysis of process executions, including performance, conformance, and variability.

*Process simulation* allows us to estimate how changes to a process (*e.g.* removing an activity) will affect the performance of process, with respect to temporal, cost, and other metrics. The starting point of process simulation is a BPS model (*Dumas et al., 2018*). A BPS model is a process model enriched with simulation parameters such as the inter-arrival time between two consecutive cases, the processing times of each activity, the number of resources available, their working timetables, and the allocation of resources to task, *etc.* When simulated, a BPS model produces a *simulated log*, consisting of a set of activity instances. From this simulated log, we can extract various performance measures, such as processing and waiting times of each activity, execution costs, and resource workload.

Alternative process configurations, known as what-if scenarios, can be created by adjusting simulation parameters or modifying the control-flow of the process model relative to a baseline BPS model (AS-IS BPS). Changes to the control-flow level must be approached carefully to avoid introducing deadlock issues. Typically, these types of changes adhere to adaptation patterns outlined by *Weber, Rinderle & Reichert (2013)*, which provide guidelines for modifying the process model based on specific requirements. These adjustments may involve inserting, deleting, parallelizing, or replacing process fragments.

*Data-driven simulation* uses PM techniques to automatically discover BPS models that closely mirror actual process executions (*Khodyrev & Popova, 2014*; *Pourbafrani, van Zelst & van der Aalst, 2020*; *Camargo, Dumas & González-Rojas, 2020*). The most commonly used techniques to discover process models rely on stochastic process models. While these stochastic models effectively capture the control-flow of the process and the stochastic branching behavior at decision points, they do not incorporate other aspects captured in a simulation model such as resource capacity and activity processing times.

### Generative process models

A DL model connects layers of neurons, also known as perceptrons, which collectively perform non-linear data transformations to learn patterns observed in the data (*LeCun, Bengio & Hinton, 2015*). Various neural network architectures have been proposed, including feed-forward networks, convolutional neural networks (CNN), variational auto-encoders, generative adversarial networks (GAN), and recurrent neural networks (RNN). Some of these architectures have been applied for predictive process monitoring.

*Evermann, Rehse & Fettke (2017)* proposed an RNN architecture to generate the most likely remaining sequence of events in an ongoing process execution (a case). This approach, along with the one presented in *Lin, Wen & Wang (2019)* and others reviewed in *Tax, Teinemaa & van Zelst (2020)*, lack the capability to handle numeric features and thus cannot generate timestamped events. *Tax et al. (2017)* use an RNN architecture known as Long-Short-Term Memory (LSTM) to predict the next event and its timestamp in an ongoing case, and to generate the remaining sequence of events. This approach faces limitations in handling high-dimensional inputs due to the use of one-hot encoding of categorical features, what decreases the accuracy as the number of categorical features increases. The DeepGenerator approach (*Camargo, Dumas & González-Rojas, 2019*) employs a RNN architecture to generate long suffixes and entire traces, by using a random next-event selection approach. It associates timestamps and resources to each event in a trace, and also incorporates two mechanisms to handle high-dimensional inputs: n-grams and embeddings. The authors in *Di Mauro, Appice & Basile (2019)* propose the use of stacked inception CNN modules for the next-activity prediction problem. They demonstrate that this neural network architecture outperforms RNN architectures in computational efficiency and prediction accuracy across various real-world datasets. However, this approach cannot generate long suffixes and entire traces. *Taymouri et al. (2020)* proposed a GAN-LSTM architecture to train generative models that produce timestamped activity sequences, but without associating resources with each event. *Camargo, Dumas & González-Rojas (2022)* introduced the DeepSimulator method for learning generative models for business process simulation by combining techniques from both DDS and DL models (specifically, long short term memory (LSTM)-based architectures). This approach aims to leverage the accuracy and precision of DL models in generating start and end times of activity instances, while retaining the characteristics of DDS models to perform what-if analysis.

RNNs are specifically designed to handle sequential data pattern recognition tasks. Among RNN architectures, LSTMs are notable to capture temporal sequences, contextual attributes, and their long-range dependencies more accurately compared to conventional RNNs (*Sak, Senior & Beaufays, 2014*). LSTM networks have demonstrated their suitable for generating complete traces of event logs (*Camargo, Dumas & González-Rojas, 2022*). This capability is particularly useful in simulation scenarios, as it enables the generation of hypothetical behaviors that closely mimic the observed behaviors in the original log. We used and extended the LSTM architecture from the DeepGenerator to generate complete traces of an event log that comply with the user-specified control-flow changes in a declarative manner.

### Constraints specification on process models

Declarative specifications in business process modeling involves defining behavioral rules that govern the permissible executions of a process. The key idea of declarative process specifications is that, in the absence of any specification, any sequence of activities is allowed within a process. The boundaries of the process (*i.e.,* what is not allowed) is defined by means of constraints that specify the conditions under which activities must, can, or cannot be executed, often influenced by the occurrence or absence of other activities.

DECLARE, a declarative process specification language, offers a set of templates for formulating constraints (*Di Ciccio & Montali, 2022*). In this article we employ DECLARE as the modeling language of changes, leveraging its extensive array of available tools and techniques. In our approach, we use a subset of DECLARE constraints to express control-flow changes in a declarative manner. Subsequently, we automatically generate what-if models adapted to the new execution dynamics.

## Related work

Related work discusses DDS approaches and methods for predicting or generating process executions from restrictions.

### Data-driven simulation of business processes

A key challenge on DDS methods lies in the capability to derive, validate, and tune BPS parameters from event logs. The tuning phase is crucial for optimizing the similarity between the original log and the logs produced by the extracted BPS model. Several studies have addressed this challenge through various approaches.

Some studies have proposed conceptual frameworks and guidelines to manually extract BPS parameters (*Wynn et al., 2008*; *Martin, Depaire & Caris, 2016*). Others automate the extraction of simulation parameters from the event logs, which is particularly beneficial for easing the creation of simulation scenarios. For instance, *Rozinat, Mans & van der Aalst (2009)* proposed a semi-automated approach based on Colored Petri Nets, while *Khodyrev & Popova (2014)* introduced an approach to extract BPS models from data although leave aside the discovery of resource pools who perform activities. Similarly, *Pourbafrani, van Zelst & van der Aalst (2020)* proposed a method for generating DDS models based on time-aware process trees.

More recently, *Mannhardt et al. (2023)* and *de Leoni et al. (2023)* proposed a method for discovering simulation models wherein the current state of a case is determined both by the activities that have previously occurred in the case, and the values of one or more variables associated to the process. The authors show that this approach improves the accuracy of the sequences of activities generated by the simulator.

However, while the above approaches tackle the problem of discovering an as-is simulation model, they do not address the problem of how to enable users to specify what-if analysis scenarios.

In our approach, we use Simod (*Camargo, Dumas & González-Rojas, 2020*) as a DDS method due to its fully automated capability to discover and tune BPS models from event logs (see 'Evaluation'). Simod employs Bayesian optimization to tune the hyperparameters

used to discover the process model, resource pools, and statistical parameters of the BPS model, including branching probabilities, processing times of activities, and inter-arrival times among cases.

While existing approaches effectively generate baseline (AS-IS) BPS models from process executions, they fall short in automating the configuration of target simulation models (what-if scenarios) based on user-defined constraints. This gap forces analysts to manually adjust the control-flow relations in the discovered simulation models to specify what-if scenarios, such as re-sequencing tasks. However, this task is challenging and error-prone due to the complexity of DDS approaches, which often produce simulation models with numerous control-flow transitions and complex branching and looping structures. Additionally, altering these relations manually may inadvertently introduce control-flow structures that lead to deadlocks. Although *Golfarelli & Rizzi (2010)* presents a methodology for designing what-if analysis at a conceptual level, the practical implementation of this modeling approach remains highly complex and challenging if performed manually.

Recent DDS studies, including *López-Pintado et al. (2021)* and *Bejarano et al. (2023)*, delve into the use of simulation scenarios discovered by DDS approaches to optimize the allocation of resources to activities in a process. *López-Pintado et al. (2021)* propose a multi-objective optimization approach aimed at searching for optimal resource allocations with respect to temporal and cost performance measures. Their approach takes an as-is simulation model (discovered *via* DDS techniques) and perturbs them to generate a number of what-if scenarios to explore the performance of different resource allocations. Similarly, *Bejarano et al. (2023)* introduce an approach to automate the discovery of optimal resource allocations. In their method, what-if scenarios are generated based on user-defined preferences and allocation policies. Even though these approaches automate the creation of what-if scenarios by adjusting parameters of the simulation model, they do not address control-flow changes. Instead, the focus of these approaches is on altering the allocation of resources to activities in the process. Control-flow changes alter the structure of the process, making it challenging to create correct (deadlock-free) simulation models.

### Generation of process executions from restrictions

The authors in *Di Francescomarino et al. (2017)* use a variant of linear temporal logic (LTL), tailored for finite traces, to express a-priori knowledge regarding the future of ongoing process executions. LTL rules are constructed using four temporal operators (*i.e.,* next, future, globally, until) to delineate the anticipated development of an ongoing case. This approach improves prediction accuracy by discarding predictions(sequences) that do not adhere to the defined LTL rules. In contrast to this approach, our investigation focuses on generating traces or suffixes labeled not only with sequences of activities but also with timestamps and resources. This enables predictions to be used within the context of temporal analysis within what-if scenarios. Achieving this requires a sophisticated approach to model and combine the generated traces with traces discovered from past executions.

In this research, we assume that process monitoring approaches, which predict the sequence of future activities entirely from past executions, serve as input for our proposed approach. Our goal is to introduce a novel literature-first approach for automating the

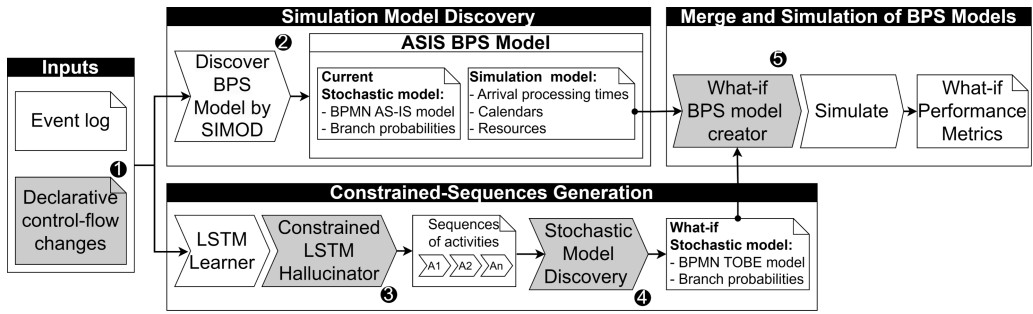

**Figure 2** Overview of the proposed approach to create BPS models from declarative changes.

creation of what-if scenarios from declarative control-flow changes within the realm of business process simulation. Consequently, the proposed method enables business users to evaluate the performance implications of such changes in a simulated environment.

Business processes often display a combination of two distinct characteristics. Less structured processes with a high level of variability can be aptly described using declarative languages like DECLARE, providing a compact representation. Conversely, more stable processes with well-structured control flows are better represented using procedural languages such as Petri Nets, enabling a more detailed and granular modeling approach. The choice between declarative and procedural languages depends on the nature of the process, with declarative languages preferred for capturing variability and offering concise descriptions, while procedural languages excel in representing structured control flows and providing more detailed models (*De Masellis et al., 2017*). *De Masellis et al. (2017)* introduce a technique that automatically adapts procedural models to adhere to sets of declarative rules. The authors present a theoretical characterization and propose an automata-based solution. They extensively tested their approach against state-of-the-art techniques for process discovery and model repair (*De Masellis et al., 2017*). While the results demonstrate the feasibility of adapting declarative rules to procedural models, this problem has not been addressed in the context of creating simulation scenarios from declarative rules.

## AUTOMATIC GENERATION OF SIMULATION SCENARIOS WITH CONTROL-FLOW CHANGES

This section outlines the proposed approach for automatically generating simulation scenarios based on declarative control-flow changes. Figure 2 illustrates the primary components of the proposed pipeline, where critical steps extending existing predictive monitoring approaches are numbered and linked with the subsequent subsections.

The pipeline takes an event log and a set of rules describing control-flow changes as input to generate a simulation scenario. These control-flow changes specify the scope of the activity sequences to be generated based on user-defined constraints (see 'Control-flow change specification'). The event log is leveraged to discover the current simulation model associated with the process (see 'Simulation model discovery') and to train a generative

LSTM model using previously observed traces. These traces guide the generation of activity sequences that adhere to the specified constraints (see 'Constrained-sequences generation'). Subsequently, a what-if stochastic process model is discovered based on the generated activity sequences (see 'What-if stochastic model discovery'). In the next step, the simulation parameters discovered earlier are combined with the what-if stochastic process model to produce a simulation scenario that aligns with the specified rules (see 'Merge and simulation of BPS models'). Finally, this what-if scenario is executed in an existing process simulator, and the anticipated performance metrics are calculated for analysis regarding potential process improvements.

## Control-flow change specification

As a syntax for specifying and implementing declarative control-flow changes, we employ the structured format of constraint definitions in the DECLARE modeling language. This structural definition is intuitive and easy to comprehend. For instance, the expression *Task A* >> *Task B* denotes a sequential relationship where Task A is followed by Task B, while the expression [*Task D*, *Task E*] denotes that Task D is executed parallel to Task E.

We implemented four constraints of the DECLARE language (*Di Ciccio & Montali, 2022*) to specify different declarative control flow changes and to test its implementation in the generative approach:

1. The *Eventually follows* constraint specifies that a sequence of tasks must execute in a specific order, but they do not necessarily need to be consecutive or appear together. Tasks may occur between the specified sequence. For instance, the following syntax expresses a change where Task C eventually follows Task A, with the $*$ symbol indicating that any task can occur between tasks A and C.
   *Task A* >> $*$ >> *Task C*

2. The *Directly follows* constraint specifies a set of activities that should be performed one after another without any activities in between. For example, the following syntax expresses a change where Task B directly follows Task A.
   *Task A* >> *Task B*

3. The *Task required* constraint specifies that a specific task is mandatory and must be present in all traces of the process. For instance, the following syntax expresses a change where Task A is required.
   *Task A*

4. The *Task not allowed* constraint specifies that a particular task should be excluded and not appear in any process instance. For example, the ˆsymbol denotes that Task A is not allowed in the generated sequences.
   ˆTask A

The definition of what-if scenarios involves specifying two distinct types of control-flow changes: restrictive changes and non-restrictive changes, also named as enhancing changes.

**Definition 1 (restrictive changes).** *Given a process model discovered from an event log, a restrictive change involve modifications to the control-flow behaviors observed within the event log that comply with the specified change. These modifications are applied exclusively to the traces existing within the event log, restricting the process model's behavior to align with*

*the defined constraints. Traces that do not meet the specified change remain unaffected by the restrictive modification.*

Figure 3 illustrates how the control-flow of the process depicted in Fig. 1 is adjusted to adhere to restrictive changes based on the four aforementioned rules. For instance, in the scenario where Task E is deemed mandatory, the OR gate in the baseline model is replaced by an AND gateway to ensure that all process executions include Task E. The original model allowed for traces where either Task E or Task D were executed, or both tasks were executed in parallel.

**Definition 2 (enhancing changes):** *Given a process model discovered from an event log, a non-restrictive change or enhancing change involves the addition of control-flow behaviors not observed in any trace within the event log.*

Figure 4 illustrates how the control flow of the process presented in Fig. 1 can change to comply with two non-restrictive path rules. For instance, when specifying that Task C directly follows Task A, the order of tasks B and C are exchanged. This behavior is different from the behavior observed in the original process model since there were no traces within the event log satisfying this condition.

The running example was employed to illustrate the types of changes due to its straightforward structure. Changing manually the control flow of a typical and real-life BPS model would be very complex and difficult to ensure its consistency. This complexity is illustrated in the validation section.

## Simulation model discovery

We use Simod (*Camargo, Dumas & González-Rojas, 2020*) as a tool to discover a complete BPS model from an event log. Simod uses Bayesian optimization to tune the hyperparameters used to discover the process model, the resource pools, and the statistical parameters of the BPS model (branching probabilities, activity processing times, inter-case arrival times, and distributions). This tuning seeks to optimize the similarity between the logs produced by the discovered BPS model and the original log.

## Constrained-sequences generation

We adapted the DeepGenerator method (*Camargo, Dumas & González-Rojas, 2019*), a technique that utilizes neural networks with the LSTM architecture to predict sequences of process events. We introduced a new filtering phase that exclusively incorporates traces conforming to user-specified constraints. The created hallucinator generates complete event traces along with their corresponding attributes (timestamps, resources) that adhere to the control-flow changes specified by the analysts. Consequently, it generates new traces distinct from the original event log traces used to train the LSTM model. For example, we allow the specification that activity A follows activity B in a sequence. This capability facilitates the generation of traces that not only resemble the original log traces used for training but also adhere to the new constraints. The constraint-based trace generator consists of two phases.

In the initial phase, a deep learning model is trained using the original event log. To this end, DeepGenerator feeds an LSTM network with feature vectors extracted from each

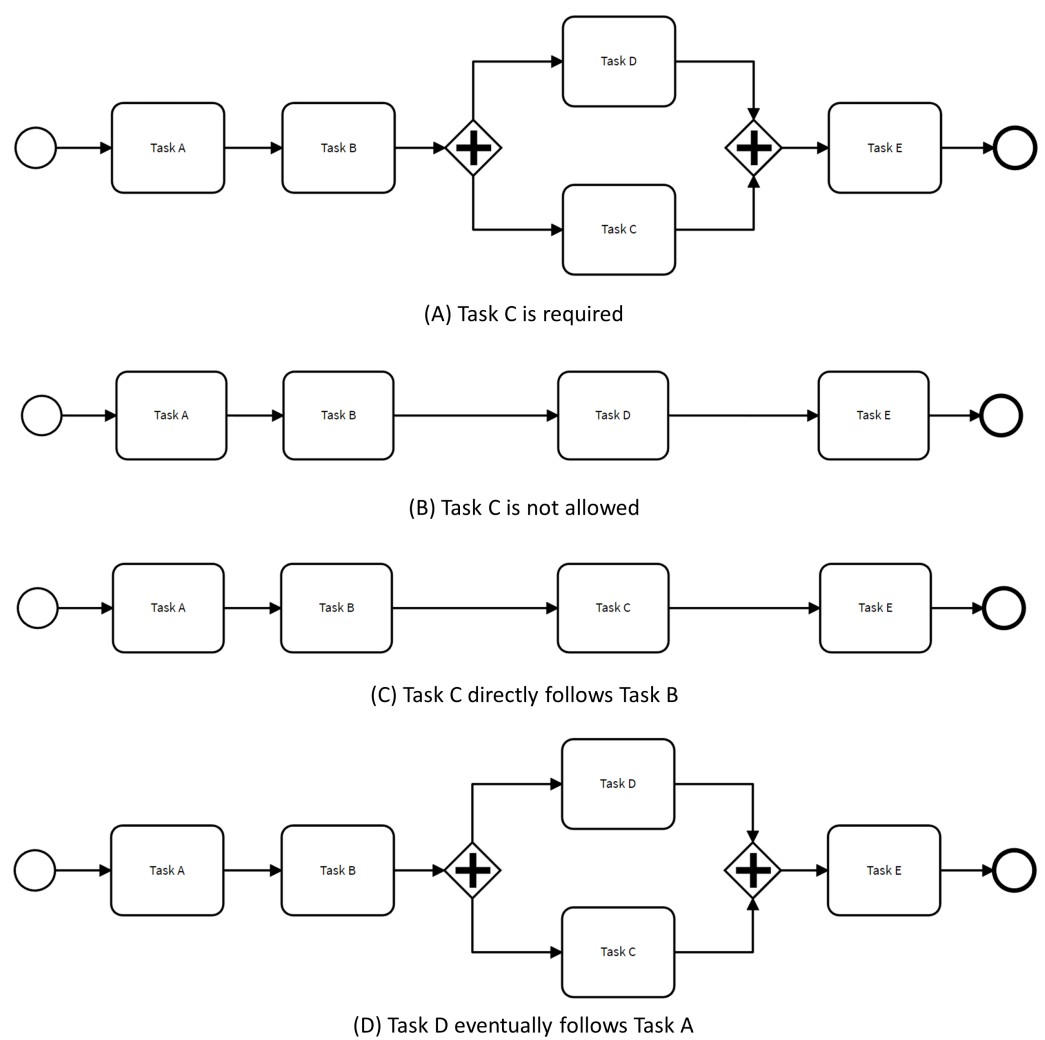

(A) Task C is required

(B) Task C is not allowed

(C) Task C directly follows Task B

(D) Task D eventually follows Task A

**Figure 3** **Process models complying with restrictive changes.**

prefix of a trace in the training set. Each prefix is encoded by means of a feature vector that contains information about the activities, the resources, and the relative times (*i.e.* the durations since the start time of the case). Specifically, given a prefix of a trace of length $k$, three $n$-grams are extracted from the prefix ($n$ being a hyper-parameter): one $n$-gram for the activity, one for the resource, and one for the relative time. The $n$-gram of a prefix is either:

- If $n \leq k$: $n - k$ null values (represented by zeroes), followed by the most recent $n$ activities, resources, or relative times in the prefix;
- or $n > k$ each of the most recent $n$ activities, resources, or relative times in the prefix;

For example, given the trace $\sigma = [(e_1, r_1, 3), (e_2, r_2, 7), (e_3, r_3.12), (e_4, r_4, 18), (e_5, r_5, 20), (e_6, r_6, 24)]$, hyper-parameter $n = 4$, and prefix length $k = 2$, the prefix of $\sigma$ of length $k$, is encoded *via* three 4-grams: $\langle 0, 0, e_1, e_2, e3, e_4 \rangle$ , $\langle 0, 0, r_1, r_2, r_3, r_4 \rangle$, and $\langle 0, 0, 3, 7, 12, 18 \rangle$.

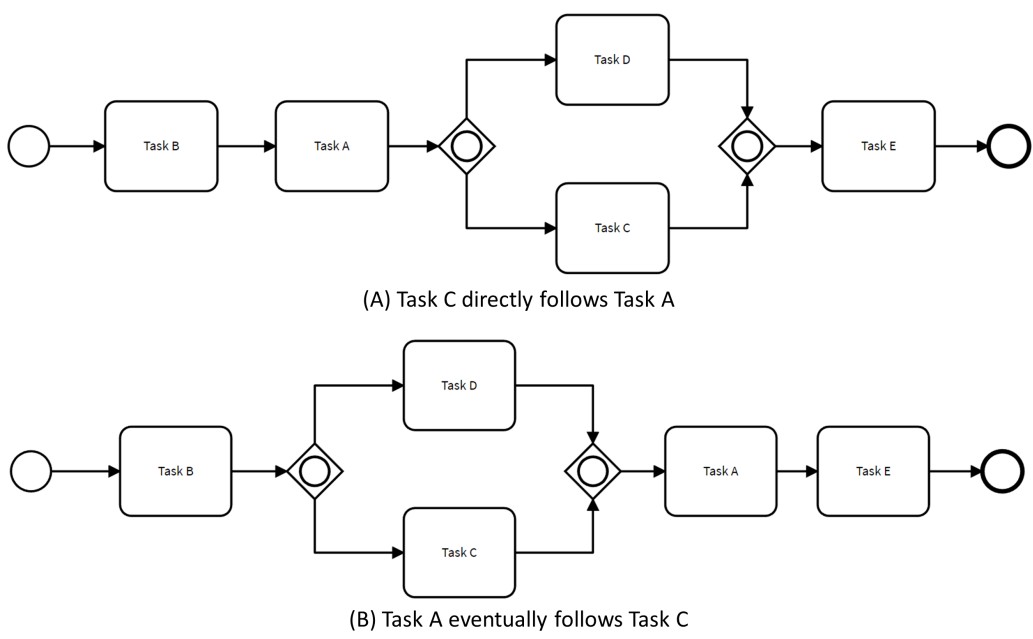

(A) Task C directly follows Task A

(B) Task A eventually follows Task C

**Figure 4** **Process models complying enhancing changes.**

LSTMs are not suited to handling categorical attributes with a large number of possible values. In real-life scenarios, the number of activity types may be in the order of hundreds and the number of resources may be in the order of hundreds or thousands. Accordingly, we use neural network embeddings as an approach to reduce the dimensionality brought about by the presence of hundreds of activities and/or resources, resources are grouped in roles. Embeddings map each symbol (in our context, each activity type) into a numerical feature vector.

For training purposes, the network architecture consists of an input layer for each of the three features, two stacked LSTM layers with 100 neurons each, and a dense output layer. Specifically, the inputs describing categorical variables (activities and roles) are concatenated and shared with the first LSTM layer, while the inputs for discrete variables (relative times) are independently shared with the first LSTM layer. This separation of features with different natures (*i.e.,* categorical or continuous) helps prevent potential noise and enables the distinction of execution patterns when exchanging information between LSTM layers. Additionally, a batch normalization LSTM layer is defined for each feature utilized. Finally, an output layer is designated for each feature, with a size equivalent to the number of activities, roles, or time bins. *Camargo, Dumas & González-Rojas (2019)* demonstrated that the most effective model architecture was the shared categorical approach described above, achieving optimal performance across various datasets.

In the second phase, traces are generated sequentially from the trained model. The approach generates a comparable number of traces to the original event log used for training the LSTM model. By exclusively generating traces that adhere to the specified

constraints, the distribution of traces changes, thereby affecting the frequency of activity occurrences within the generated event log. However, to accommodate these changes in activity distribution, a what-if stochastic process model is derived from the generated event log. This model predicts subsequent activities following an initial activity, facilitating the generation of traces in the generated event log. Importantly, only traces that conform to the proposed restrictions are included in the generated log, ensuring compliance with the specified constraints. It is crucial to clarify that the hallucinator model does not attempt to modify the branching probabilities at decision points within the model. Instead, it learns from the behavior of the historical traces on which it was trained, generating distinct traces based on the acquired knowledge. This phase comprises two stages: post-processing and filtering traces. In the post-processing stage, two methods are implemented for predicting new events: selecting the most probable event based on the model output or randomly choosing an activity to introduce diversity in the generated traces. During a filtering stage, the trace constructed from the preceding stages undergoes evaluation to determine if it adheres to the specified constraint. If the trace meets the imposed restriction, it is included in the predicted event log; however, if it fails to comply with the constraint, it is excluded and not incorporated into the predicted event log.

The generation of traces for what-if purposes involves different strategies for training and prediction. Firstly, when the available event log is small, it becomes advantageous to train a generative model using the available log and subsequently generate new traces that adhere to the desired condition. This approach enables the generation of behaviors that may not be present within the traces of the original event log from which the generative model was trained. Secondly, when the available event log is large, comprising a substantial number of traces, it is more feasible to filter the traces that meet the specified condition directly from the log. However, it is crucial to emphasize that only rules that are explicitly contained within the traces of the original log can be defined and applied. These considerations are significant when implementing a solution. The proposed approach assumes that not all event logs used contain a sufficient number of traces for effective filtering. Furthermore, there is a desire to have the capability to generate traces exhibiting behaviors that are not explicitly captured within the traces of the original event log.

## What-if stochastic model discovery

The next step in the proposed method is to discover a stochastic process model of the "what-if" process, based on the set of generated traces that adhere with the control-flow changes specified by the user. A stochastic process model is a process model where every control-flow transition coming out of a decision gateway (a.k.a. *conditional flows* in BPMN) is associated with a probability. The BPMN model associated with the stochastic process model can be automatically discovered from the set of generated traces by using any PM technique. In our implementation of the proposed method, we use the Split Miner (*Augusto et al., 2019a*) algorithm because it has shown to achieve high levels of accuracy, measured *via* fitness and precision measures (*Augusto et al., 2019b*). Additionally, it ensures that the discovered process model is deadlock-free. Alternatively, we could use another algorithm

with similar properties, such as Inductive Miner algorithm or its variations (*van Detten, Schumacher & Leemans, 2023*).

Once the BPMN model is available, each trace in the generated log is replayed against the process model using a trace alignment technique. Trace alignment techniques enable us to determine how to replay a trace against a process model even when the model cannot perfectly parse the trace (*e.g.*, because an activity in the trace or in the model needs to be skipped to complete the replay). There are numerous techniques for trace alignment (*Carmona, van Dongen & Weidlich, 2022*). In our implementation of the method, we utilize the trace alignment technique proposed in *Reißner et al. (2019)* due to its computational efficiency in practical scenarios. A replay algorithm based on trace alignment allows us to determine which elements in the BPMN model (*e.g.*, flow relations) are "traversed" by a given trace in a log. While replaying traces against the discovered BPMN model, we calculate the number of times each conditional flow is traversed. This process yields a traversal frequency metric for each conditional flow. The resulting traversal frequencies are then normalized to obtain a branching probability for each conditional flow in the model, which allows us to derive a stochastic process model.

### Merge and simulation of BPS models

Once the What-if stochastic process model is generated, it is combined with the parameters discovered in the AS-IS BPS model to create a What-if BPS model. This model defines a target simulation scenario that incorporates information from the original log and reflects the behavior of the traces as defined by the constraints. Subsequently, the What-if BPS model is executed into an existing discrete-event process simulator to analyze the expected performance metrics, such as cycle time and resource utilization. These metrics offer insights into how the proposed change impacts the model's performance, facilitating decision-making for improvement, whether through manual analysis or automated evaluation based on predefined criteria.

## EVALUATION

We empirically compare the BPS models generated by Dynamics to adhere with declarative control-flow changes with BPS models created manually in terms of the similarity of the simulated logs they generate. This article addresses the following research question: *To what extent can the proposed approach replicate the effect of restrictive and enhancing control-flow changes on the performance metrics of a process?*.

This section describes two experimental evaluations to assess the effectiveness of the proposed approach. The first experiment compares the effect of both restrictive and enhancing changes on synthetic logs. Through this analysis, we aim to assess how these types of changes, corresponding to the four constraints incorporated in the approach, influence the behavior and performance of simulated process instances. The second experiment compares the effect of restrictive changes on a real-life event log, which contains complex control-flow behaviors. By conducting this experiment, we seek to gain valuable insights into the applicability and effectiveness of the approach in real-world scenarios.

**Table 1   Event logs description.**

| Source | Event log | Traces | Variants | Activities | Resources |
|--------|-----------|--------|----------|------------|-----------|
| Real | CDM | 954 | 94 | 16 | 559 |
| Synthetic | PE | 608 | 70 | 21 | 47 |
| Synthetic | RE | 540 | 3 | 7 | 50 |

## Datasets

We selected one real event log and two synthetic event logs to evaluate the accuracy of the proposed approach in replicating the estimation of impact due to control-flow changes. These logs exhibit different characteristics in terms of control-flow, execution times, and resources. Resources, in particular, have a significant impact on process performance since they carry out costs, time, and quality. Table 1 characterizes these logs.

The Running Example (RE) log was created to provide a scoped explanation of the different types of changes defined in the proposed approach. The Purchasing Example (PE) corresponds to the synthetic purchase-to-pay process event log, which is publicly accessible (http://fluxicon.com/academic/material/). The Consulta Data Mining (CDM) event log contains the execution data of the academic credentials recognition process from a Colombian University. This log was generated from its business process management (BPM) system. The CDM log contains complex control-flow behaviors, making it challenging to manually incorporate changes.

The various components of the proposed tool's pipeline were implemented as Python packages. Table 2 provides an overview of the parameters used to run SIMOD for discovering the AS-IS BPS models. To discover the What-if stochastic process models, we used the following parameters: a parallelism threshold of 50% (epsilon), and a percentile for frequency threshold of 70% (eta). The epsilon parameter determines the number of concurrent relations captured between events, whereas the eta parameter specifies the most frequent paths between activities within the top $\eta$ percentiles.

Table 3 provides an overview of the characteristics of the AS-IS BPS models discovered by Simod from the event logs. These BPS models contain a subset of the original traces due to the hyperparameters used by Simod.

Simulation parameters related to resources were configured as follows: the CDM event log contains 347 resources grouped into ten roles, the PE event log comprises 47 resources grouped into eleven roles, and the RE event log includes 50 resources grouped into two roles. Each role across all BPS models has an standard cost of 20 EUR per hour. Simulation parameters related to gateways include probabilities discovered for the six split gateways of the CDM event log, the three gateways of the PE event log, and the single gateway of the RE event log. Simulation parameters related to inter arrival times shows an exponential inter arrival time with a mean of 99.9 min for the CDM event log. The PE event log also exhibits an exponential inter arrival time with a mean of 79.8 min. In contrast, the RE event log follows a normal inter arrival time with a mean of 10 min and a standard deviation of 1.4 min. The average and maximum process duration measures presented for each AS-IS

**Table 2  SIMOD parameters used to discover AS-IS BPS models.**

| Parameter | Value |
|---|---|
| repetitions | 10 |
| simulation | True |
| sim_metric | tsd |
| add_metrics | ['day_hour_emd', 'log_mae', 'dl', 'mae'] |
| concurrency | 0.0 |
| epsilon | 0.5 |
| eta | 0.7 |
| alg_manag | replacement |
| gate_management | discovery |
| rp_similarity | 0.8 |
| res_cal_met | discovered |
| arr_cal_met | discovered |
| res_dtype | 247 |
| arr_dtype | 247 |
| pdef_method | automatic |
| res_support | 0.05 |
| res_confidence | 70 |
| arr_support | 0.05 |
| arr_confidence | 10 |

**Table 3  Characteristics of the discovered AS-IS BPS models.**

| BPS model | Traces | Activities | Resources | Gates | Avg. Duration | Max. Duration |
|---|---|---|---|---|---|---|
| CDM | 57 | 8 | 347 | 6 | 3.5 days | 9.8 days |
| PE | 32 | 20 | 47 | 3 | 496.3 days | 759.5 days |
| RE | 35 | 5 | 50 | 1 | 2.8 days | 23.1 days |

BPS model serve as reference points to contrast with the simulation results of the what-if scenarios created in the experiments.

## Evaluation measures

To evaluate the accuracy of BPS models generated by Dynamics, we compute the distance rate (error rate) of cycle times between a BPS model generated from declarative changes and a manually created BPS model (ground-truth). The distance measures the temporal similarity between two logs generated by the simulation. The distance of a pair of models is calculated as the ratio of their mean cycle times, represented by a discrete value from 0 to 1.

Process simulators generate execution logs from BPS models and expose their expected performance in terms of traditional metrics such as cycle time and cost. The *process cycle time* measure denotes the overall duration necessary for the completion of the entire process, including the time taken for each activity and any waiting periods between activities. It helps to determine the efficiency of process by calculating the ratio between the time spent executing the activities and the overall cycle time. The *cost* measure represents

the total expenses involved in completing the process, encompassing both the average time taken for activity execution and unit cost per hour associated with the resources allocated to the tasks.

While our comparison focuses on the distance of cycle times, it is important keep in mind that control-flow changes can significantly affect cost performance. Measuring cost independently from cycle time helps to scope the impact on what-if scenarios from different dimensions. Changing the order tasks are executed could generate greater waiting times and therefore greater costs. Eliminating tasks will change the process execution impact on cost depending on the resource that is allocated to the task since each resource can have a different unit cost per hour.

In both experiments, the reported results represent average cycle times that include off-timetable hours from multiple executions (cases) of each process simulation. The simulation parameters—such as arrival rate, inter-arrival times, time constraints on activities, resources allocation, and timetables—remain consistent for baseline and target BPS models. This approach ensures a fair comparison and allows for a direct assessment of the impact of control-flow changes on process performance. By maintaining consistent simulation parameters across both models, we can isolate the effects of the control-flow changes and evaluate the performance differences between the baseline and target scenarios.

### Experiment 1: what-if generated models for simple logs
#### Setup
Synthetic datasets were utilized in the first experiment, where a total of six distinct changes were implemented. These changes correspond to the different rules and types of changes proposed in the approach. The distribution of these changes is as follows: one change for the "is required" rule with a restrictive change, one change for the "is not allowed" rule with a restrictive change, one change for the "eventually follows" task with a restrictive change, one change for the "eventually follows" task with an enhancing change, one change for the "directly follows" task with a restrictive change and one change for the "directly follows" task with an enhancing change.

Two BPS models were created from each of these changes: a What-if manual model and a What-if generated model. The changes to BPS models created manually were implemented using the Apromore PM tool, and its corresponding BPS model was discovered with Simod. This BPS model and its estimated performance metrics are assumed as the ground-truth. The changes to BPS-generated models were specified as declarative rules. Simod was also used to obtain identical simulation parameters for both the manually generated simulation models and the input simulation models used in the proposed tool. This approach ensures the comparability of the results obtained from the two sets of models.

#### Results
Table 4 presents the results of experiment 1. The results indicate that the BPS models generated from declarative specifications and the manually modified BPS models yield similar performance metric values. Both generated and manual what-if BPS models exhibit process durations that are consistent with the process duration of the AS-IS BPS model (see Table 3).

**Table 4  Comparison of performance metrics between generated and manually created BPS models.**

| Event log | Type of change | Rule | Average process duration | | |
|---|---|---|---|---|---|
| | | | Generated | Manual | Error |
| PE | Enhancing | Choose best option >> Settle Conditions With Supplier | 241.81 days | 495.13 days | 0.51 |
| PE | Enhancing | Settle Conditions With Supplier >> * >> Create Purchase Order | 319.66 days | 406.61 days | 0.21 |
| PE | Restrictive | ^Ammend Request for Quotation | 204.96 days | 452.82 days | 0.54 |
| PE | Restrictive | Create Request for Quotation >> * >> Create Purchase Requisition | 246.61 days | 500.18 days | 0.50 |
| PE | Restrictive | Send Request for Quotation to Supplier >> Analyze Request for Quotation | 254.12 days | 442.16 days | 0.42 |
| PE | Restrictive | Settle Dispute With Supplier | 366.41 days | 499.61 days | 0.26 |
| RE | Enhancing | Task C >> * >> Task A | 2.57 days | 2.87 days | 0.10 |
| RE | Enhancing | Task A >> Task C | 1.95 days | 2.88 days | 0.32 |
| RE | Restrictive | Task B >> Task C | 3.11 days | 2.60 days | 0.19 |
| RE | Restrictive | ^Task C | 1.88 days | 1.94 days | 0.02 |
| RE | Restrictive | Task C | 2.29 days | 2.29 days | 0.00 |
| RE | Restrictive | Task A >> * >> Task D | 1.92 days | 2.62 days | 0.26 |

The results for the Purchasing Example (PE) synthetic log exhibits average process durations of the generated BPS models lower than those of the manual models. A possible reason is that the structure of the generated models incorporate multiple gateways, what creates multiple execution paths decreasing the average execution time. The cycle time metric exhibits similar results for the restrictive and enhancing changes (39% average error). The results for the Running Example (RE) synthetic log exhibit greater similarities between generated and manual BPS models. The process duration performance metric presents similar error results for both types of change (14% average error). This could occur in this synthetic log since there was no enough data to observe and train the models, therefore all changes are assumed as enhancing since their involved behaviors were not observed in the original event log.

## Experiment 2: what-if generated models for a real-life log
### Setup
The CDM dataset was employed in a second experiment to discuss the complexity of manually implementing changes in real-life processes with large traces and variants. Figure 5 illustrates the complexity for changing manually the control flow within a process with large variants. This process model contains 954 cases, 94 case variants, 13.700 events, and 16 activities to be analyzed at the moment of applying a change. For example, attempting to manually implement the change where "Validacion final" (*Task A*) directly follows "Validar solicitud" (*Task B*) proves to be an intricate task. This change requires deleting all the arcs where this condition is not met. In particular, the arcs between activities such as "Revisar curso" (*Task C*), "Homologacion por grupo de cursos" (*Task D*), "Visto

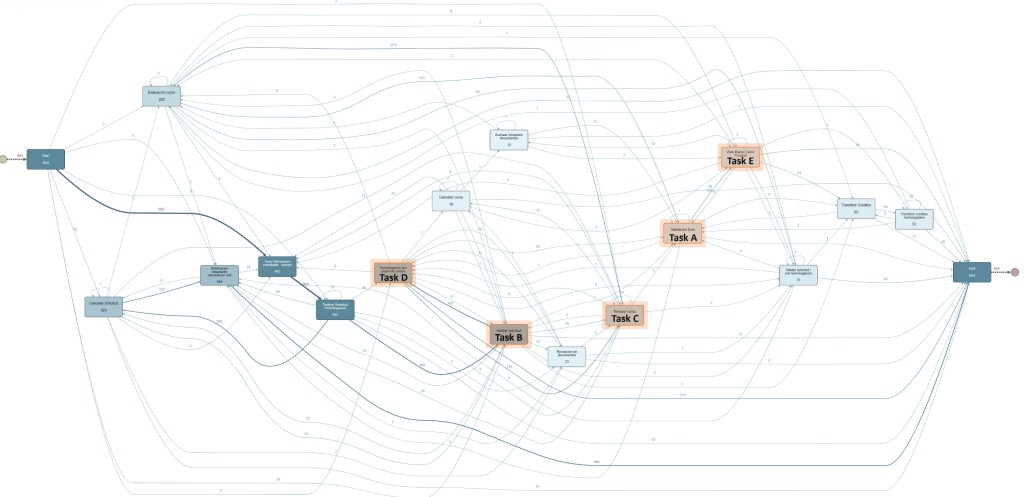

**Figure 5** **CDM process model discovered from the event log.**

bueno cierre proceso" (*Task E*) to the activity *Task A* must be deleted whereas warranting the control-flow consistency in all involved gateways.

Due to the complexity of applying manual changes to large process models without creating border effects, we had to apply an extra step by manually filtering the AS-IS process model in the Apromore PM tool to retain the traces complying with the specified change constraints. Therefore, the BPS model discovered from the filtered log and its simulated performance metrics become the ground-truth. Consequently, only restrictive changes were implemented for comparing the similarity of performance metrics against the generated BPS models. The distribution of the changes is as follows: one change for the "is required" rule, one change for the "is not allowed" rule, one change for the "directly follows" rule, and one change for the "eventually follows" rule.

Two BPS models were created from each of the restrictive changes: a What-if filtered model and a What-if generated model. Simod was used to discover the process model and simulation parameters of the filtered log. Similar to the first experiment, consistent simulation parameters were utilized across the various process models with the proposed changes. Additionally, for comparability purposes, the branching probabilities of the what-if process model were computed based on the filtered log instead of being computed based on the original log. This approach ensured the comparability of performance metrics across the models.

Table 5 illustrates the description of the *Consulta Data Mining* log filtered to comply with the aforementioned rules. The filtered log contains significantly less information compared to the original log. Consequently, the "res_confidence" parameter in the Simod tool had to be reduced since no calendar timetable was discovered. The remaining parameters used for this experiment were consistent with those employed in the first experiment.

**Table 5 Characterization of the manually filtered event log to comply with control-flow changes.**

| Type of change | Rule | Traces | Variants | Activities | Resources |
|---|---|---|---|---|---|
| Restrictive | *Evaluacion curso* | 229 | 54 | 16 | 258 |
| Restrictive | *Radicar Solicitud Homologacion >> * >> Visto Bueno Cierre Proceso* | 143 | 38 | 13 | 179 |
| Restrictive | *Radicar Solicitud Homologacion >> Validar solicitud* | 481 | 59 | 16 | 414 |
| Restrictive | *^Cancelar curso* | 935 | 89 | 15 | |

**Table 6 Comparison of performance metrics between BPS models generated from declarative rules and BPS models manually filtered and configured.**

| Event log | Type of change | Rule | Average process duration | | |
|---|---|---|---|---|---|
| | | | Generated | Filtered | Error |
| CDM | Restrictive | ^Cancelar curso | 9.49 days | 7.13 days | 0.33 |
| CDM | Restrictive | Evaluacion curso | 2.72 days | 3.53 days | 0.23 |
| CDM | Restrictive | Radicar Solicitud Homologacion >> * >> Visto Bueno Cierre Proceso | 8.46 days | 9.33 days | 0.09 |
| CDM | Restrictive | Radicar Solicitud Homologacion >> Validar solicitud | 2.03 days | 3.97 days | 0.49 |

## Results

Table 6 presents the results obtained in experiment 2. The error percentage in the Consulta Data Mining real-life log is low (28% average error) and similar to the obtained from synthetic logs. Both generated and manual what-if BPS models exhibit process durations that are consistent with the process duration of the AS-IS BPS model (see Table 3).

The main reason behind this error rate is that when filtering traces based on the defined constraints, the available information for building the process model and discovering simulation parameters using Simod is limited. However, it is worth noting that the tool considers all the behaviors observed in the original event log, not just those observed in the filtered log. As a result, there is a possibility of slight differences between the process models generated through trace filtering and the process model generated with the proposed tool. Upon analysis, differences were indeed observed, leading to higher error rates in the performance metrics.

In this experiment, it was not possible to incorporate the restrictive and enhancing control-flow changes manually due to the high variability of the BPS model. Thus, the effect of these changes on performance metrics was not measured to allow the comparison with the effect of changes automatically generated from user-defined constraints. Despite this, the results validate the viability of incorporating control-flow changes in complex models in a convenient way. Since this is the first study to tackle the challenge of generating what-if scenarios from control-flow changes, the results should be built in this direction as future work is developed.

## Threats to validity

The evaluation reported above is potentially affected by the following threats to the validity. Firstly, it is worth noting that the logs utilized for generating the simulation models with the

imposed constraints were not as extensive as initially anticipated. The level of information contained within each log plays a crucial role in ensuring the generative model's accuracy in producing trace sequences that adhere to the specified constraints. However, due to the challenges associated with manually implementing the various changes, logs with a relatively lower complexity were selected to facilitate the experimentation process.

## CONCLUSIONS AND FUTURE WORK

We developed a method to generate "what-if" process simulation scenarios from an initial simulation model discovered from an event log. This approach allows analysts to specify desired changes using declarative control-flow specifications, sparing them from manually modifying a potentially complex process model discovered from the input log. The method leverages the event log to train a generative deep learning model, which is then used to produce traces that comply with the declarative constraints. From these traces, we discover a stochastic process model and enhance it with additional simulation parameters (derived from the original log) to generate a what-if simulation scenario. The method supports both restrictive changes (removing behavior) and enhancing changes (potentially adding behavior).

To assess the tool's performance, two experiments were conducted. In the first experiment, two synthetic logs were used. Both restrictive and enhancing changes were applied to each log based on each defined rule. Manual modifications were created for each model according to the respective changes, and a comparison was made between the generated models and the manually modified models using the cycle time performance metric. The percentage of error was calculated to assess the similarity between the models. The results showed that the generated models closely resembled the manually modified models, with stronger performance observed for restrictive changes compared to enhancing changes. In the second experiment, one real log was used, and only restrictive changes were implemented. The manually modified models were created using filtered traces from the original log, and a simulation model was discovered from the filtered log. A comparison was then made between this model and the model generated by the tool using the cycle time performance metric. The findings indicated that the method's performance was less robust in this experiment due to differences between the model generated by the method and the model discovered from the filtered log.

The current method supports four types of declarative constraints: "is required", "not allowed", "eventually follows", and "directly follows". An avenue for future work is to focus on extending the method to include other declarative constraint templates defined in the DECLARE language.

The empirical evaluation demonstrated that the proposed method effectively captures the impact of restrictive control-flow changes. However, it performs less accurately with enhancing changes. To address this limitation, future research could integrate an attention layer into the deep learning model. This adjustment may improve the model's generalization capabilities, enabling it to generate traces that meet the declarative constraints even when such behaviors are not observed in the original log.

A limitation of the presented evaluation is its reliance on a coarse-grained metric such as the average cycle time to assess the accuracy of the simulation model after applying the control-flow changes. More nuanced methods have been proposed to evaluate the goodness of simulation models, including measures based on Earth Mover's Distance (EMD) or mean absolute error (MAE). Future research could focus on conducting a more comprehensive evaluation using these fine-grained metrics to provide deeper insights into model performance and accuracy.

The proposed approach does not account for potential effects on other parameters or components of the simulation model when user-defined constraints are implemented. For example, if a user wishes to remove an activity but does not specify the removal of the corresponding resource solely dedicated to that activity, the approach does not handle the removal of the resource. Addressing these side effects in constraint implementations will be explored in future work and is intended for inclusion in later versions of the proposed tool.

### Funding
This work was supported by the European Research Council (PIX project). The funders had no role in study design, data collection and analysis, decision to publish, or preparation of the manuscript.

### Grant Disclosures
The following grant information was disclosed by the authors:
The European Research Council.

### Competing Interests
The authors declare there are no competing interests.

### Author Contributions
- Daniel Barón-Espitia conceived and designed the experiments, performed the experiments, analyzed the data, performed the computation work, prepared figures and/or tables, authored or reviewed drafts of the article, and approved the final draft.
- Marlon Dumas conceived and designed the experiments, analyzed the data, prepared figures and/or tables, authored or reviewed drafts of the article, and approved the final draft.
- Oscar González-Rojas conceived and designed the experiments, analyzed the data, prepared figures and/or tables, authored or reviewed drafts of the article, and approved the final draft.

### Data Availability
The source code, datasets, generated simulation models, raw and summarized results are available at Zenodo: Barón-Espitia, D., Dumas, M., & González-Rojas, O. (2024).

Reproducibility package for "Automated generation of process simulation scenarios from declarative control-flow changes". Zenodo. https://doi.org/10.5281/zenodo.10475754

The source code is also available at GitHub: https://github.com/AdaptiveBProcess/DeclarativeProcessSimulation.

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
