# Peer review of "Automated generation of process simulation scenarios from declarative control-flow changes"

_PeerJ Computer Science, doi:10.7717/peerj-cs.2094_

## Round 0.1 · original submission · Minor Revisions

Based on the reviewers’ comments, you may resubmit the revised manuscript for further consideration. Please consider the reviewers’ comments carefully and submit a list of responses to the comments along with the revised manuscript.

**Language Note:** PeerJ staff have identified that the English language needs to be improved. When you prepare your next revision, please either (i) have a colleague who is proficient in English and familiar with the subject matter review your manuscript, or (ii) contact a professional editing service to review your manuscript. PeerJ can provide language editing services - you can contact us at [email protected] for pricing (be sure to provide your manuscript number and title). – PeerJ Staff

Reviewer 1 ·

Basic reporting

Datasets used for the evaluation have not been shared by the authors:
- I request the authors to share the running example log as well as the consulta data mining event log, or at the very least, filtered versions of these logs.
- For the publicly available purchasing example, I request the authors to add a reference link to the dataset.

Experimental design

no comment

Validity of the findings

Underlying data have not been provided by the authors of the manuscript.

Additional comments

This manuscript presents an approach to specify changes to a business process simulation model in a declarative manner. Existing approaches such as Data Driven Simulation allow users to discover business process simulation models from event logs, but end up producing overly complex models, making it difficult to capture what-if scenarios through manual alteration of the model.

- The manuscript is well organized, and I felt that the related work, context, design and definitions are well presented.
- The approach of using a generative deep learning model to produce traces that resemble those in the original log, while implementing the specified changes is novel, and well suited to this application.
- The experimental evaluation is well designed and thorough.

I am suggesting a minor revision because the datasets have not been shared. I request the authors to share the running example log as well as the consulta data mining event log, or at the very least, filtered versions of these logs. I also request them to add a reference link to the publicly available purchasing example dataset.

Reviewer 2 ·

Basic reporting

The work under review is about addressing the issue of complexity of manually specifying and tuning simulation models. The authors criticized existing Data-Driven Simulation (DDS) methods that often produce overly complex models, making it difficult for analysts to manually alter these models to capture what-if scenarios, particularly control-flow changes such as activity re-sequencing. The topic of research is aligned with the scope of the journal. One major issue is that paper did not highlights its contributions. I'd suggest to add the contributions at the end of the introduction section in the form of bullet points. Presently, the related work section is very brief. I'd recommend to expand it significantly by adding state-of-the-art work in the area. I'd suggest the authors, If possible, they should assign some name to their proposed approach.

Experimental design

Over all the experimentation and evaluation of the proposed approach is reasonable. However, presently the dataset used for the experimentation is Synthetic. Is there any possibility to evaluate the proposed approach using real dataset. Table 3 is presenting the comparison of Cost Vs Process Cycle Time however, it is not clear what does cost mean? Is it memory or what? Usually Process Cycle Time is also a cost then why authors have measured it separately.

Validity of the findings

As I stated above presently the validity has been performed using the synthetic dataset. I'd recommend authors to validate their approach on real dataset. Also if possible try to include some more DDS approaches in their comparison.

Additional comments

No

Reviewer 3 ·

Basic reporting

The paper is well organized and it present a new approach that is tested and evaluated by using adequate datset. However, I suggest the following comments:
1-Summarise in the Abstract how your approach is diffrent with related one
2-extand the first chapter in the section 2
3- Draw a table in the section 3 (where you illustrat the diffrences between your aproach and other proposed in the litterature)
4- Explain how you have choose your features from dataset.

Experimental design

no comment

Validity of the findings

no comment

---

## Round 0.2 · accepted · Accept

Congratulations, the reviewers are satisfied with the revised version of the manuscript and have recommended the acceptance decision.

Reviewer 1 ·

Basic reporting

no comment

Experimental design

no comment

Validity of the findings

no comment

Additional comments

Thank you for making the datasets and code publicly available.

Reviewer 2 ·

Basic reporting

The authors have addressed all my concerns. No more comments from my end.

Experimental design

The authors have addressed all my concerns. No more comments from my end.

Validity of the findings

The authors have addressed all my concerns. No more comments from my end.